# The “Planning Health in School” Programme (PHS-pro) to Improve Healthy Eating and Physical Activity: Design, Methodology, and Process Evaluation

**DOI:** 10.3390/nu15214543

**Published:** 2023-10-26

**Authors:** Margarida Vieira, Graça S. Carvalho

**Affiliations:** Research Centre on Child Studies, University of Minho, 4710-057 Braga, Portugal; graca@ie.uminho.pt

**Keywords:** health promotion, school-based intervention, transtheoretical model, participatory method approach, obesity prevention

## Abstract

Few interventions have successfully promoted healthy eating and active living among children with effective changes in anthropometric health outcomes. Well-designed interventions involving multiple strategies to convert the knowledge already available into action are needed for preventing childhood obesity. In this study, an educational programme called “Planning Health in School” (PHS-pro) was designed, implemented and evaluated to contribute to the prevention of obesity in childhood. The PHS-pro aimed at improving the eating behaviours and lifestyles of Portuguese grade-6 children towards healthier nutritional status. This paper describes and evaluates the PHS-pro concerning: (i) the research design within the theoretical framework grounded on “The Transtheoretical Model” and the stages of change; (ii) the educational components and the application of the participatory methodology to engage children to meet their needs, as active participants in their change process; and (iii) the process evaluation of the intervention. The implementation of the PHS-pro took into account the views and inputs of the participants for evaluating the educational components that should be considered in the designing of interventions aiming to be effective strategies. From the health promotion perspective, this study is important because it examines new approaches and pathways to effectively prevent overweight and obesity in children.

## 1. Introduction

Unhealthy eating and sedentary behaviours have been associated with excessive adiposity, which has been reflected in the increasing rates of overweight and obesity across countries, genders, ethnicities and all age groups [1,2]. Obesity has been identified as the greatest epidemic of the twenty-first century, and currently is considered the number one nutrition-related health concern for children [3,4]. The prevalence of overweight and obesity in the European childhood population has already surpassed 20% [5,6], forecasting major implications for public health and related healthcare costs in the future. Similar to many other countries, Portugal has followed the European obesity trend [7,8].

The consequences of overweight and obesity during childhood development are well documented, showing a general risk in psychological and physical health, such as depression or anxiety, low self-esteem, bullying and behaviour problems, as well as high blood pressure, dyslipidemia, hypertension, insulin resistance, in addition to other clinical consequences as asthma, type I diabetes and abnormalities of foot structure [9,10]. More recently, with the COVID-19 pandemic, obesity has emerged as one of the most prominent risk factors of severe COVID-19, increasing disease mortality, even in children and adolescents [11,12]. Furthermore, a large body of evidence indicates that obesity in youth imposes a considerable burden in adulthood, with a significantly increased risk of premature mortality with substantial and adverse long-term consequences such as cancer, cardiovascular diseases and diabetes [13,14].

Changing the course of the current situation implies taking action with effective interventions helping children to improve their behaviours and make healthier food choices, with regular physical activity. Since the release of the European Health Report in 2005 [15], with the recommendation for renewing efforts to promote the health of children, a great deal of attention has been given to preventing obesity and developing evidence-based effective interventions. Taking action to prevent obesity is one crucial step for achieving the United Nations 2030 Agenda goals, particularly the Sustainable Development Goal 3 (good health and well-being), to ensure healthy lives and promote well-being for all, whereby children are the foremost priority [16]. However, the most recent systematic review done by the Cochrane Collaboration [17] identified only a few interventions that have successfully promoted dietary and physical activity changes among children or reduced adiposity by anthropometric health outcomes. Indeed, there is a lack of well-designed long-term interventions involving multiple strategies, converting the knowledge already available into action [2,18]. To the best of our knowledge, Portuguese interventions that combine nutrition knowledge, as well as dietary and lifestyle behaviour changes, for preventing obesity during the growth and development of children with a controlled study design are scarce [19,20,21].

Therefore, to prevent overweight and obesity from rising in the Portuguese school environment, and given that preventive programmes in Portugal are lacking, an educational programme called “Planning Health in School” programme (PHS-pro) was designed to develop skills in children and guide them towards healthy behaviours, as well as encouraging them to be active participants in their own changing processes. Thus, we describe here the PHS-pro research design, methodology and approaches, as well as the process evaluation of the intervention for identifying its inherent advantages, constraints and limitations to help improve future interventions.

### 1.1. Theoretical Background

Healthy eating and daily physical activity have been established as behaviour priorities in preventing obesity. Additionally, schools are identified as the best settings in which to implement interventions to educate young people and promote healthy habits [22,23]. In this context, the statement of Carvalho [24] was inspiring to the designing of this educational programme: “knowledge is important, but for health behaviour changes, it is necessary to take into account people’s specific contexts and their own skills, to be able to switch, if they want, for healthier lifestyles”. From this perspective, schools can make the transition from knowledge into action, by providing the knowledge, but also supporting children to be able to choose healthy foods, and know how to eat properly.

However, multiple interacting factors influence the eating behaviour and the food choices of children, in an organised complex of four levels linking biology, personal behaviour and the environment [25]. Following this rationale, school-based interventions, in order to be effective, have to switch to a behavioural focus, instead of a knowledge-based focus [26]. Although knowledge is seen as one of the determinants of food choice with a positive impact on eating behaviour, it is quite a small contributor in altering such behaviours, and by itself rarely leads to behavioural change [27]. Nutrition knowledge and health recommendations of what and how to eat healthily, and why, are a resource of utmost importance, yet they are not enough to create immediate changes in behaviours, as hundreds of studies have already reported [25,28].

Interventions designed to support the behaviour change process should be theory-based, and adopt theoretical constructs, given that a growing body of evidence has already shown that they are more effective than those lacking a theoretical base [29]. In the last few decades, the most widely applied theories were the Social Cognitive Theory, the Health Belief Model and The Transtheoretical Model [30]. Among these, The Transtheoretical Model (TTM) was developed by Prochaska and DiClemente [31] and is more commonly referred to as the “stages of change” model. TTM offers an understanding of how people develop a new behaviour and improve skills and self-efficacy by assessing the stage of readiness for change, and providing a set of strategies (the processes of change) to be applied in the designing of health interventions [32].

With this in mind, TTM and its conceptual framework provide practical guidance for implementing educational programmes that promote behaviour changes in well-defined situations. Prochaska and colleagues [33] have explained the stages of change, the core constructs of the model, and how TTM-based approaches effectively change multiple health behaviour risks, including eating behaviours [34,35].

The TTM advocates the behaviour change as a process in which individuals can progress through several stages of change, and change is seen neither as linear progress nor an isolated event [35]. Therefore, the TTM provides an appropriate framework to build up this educational programme, designed with specific strategies throughout preventive actions, implemented and evaluated for expected positive changes in the eating and sedentary behaviours of children. The PHS-pro was constructed with the assumption that children would progress through the five stages identified by Prochaska and his colleagues [33]: (i) precontemplation—no intention to change behaviour or take action in the near future, with unawareness of their problems; (ii) contemplation—aware of unhealthy habits, interest and intention to change them in the next six months; (iii) preparation—intending to take action in the next 30 days; (iv) action—making visible modifications in specific behaviour targets within six months; (v) maintenance—working to continue and consolidate healthier habits. In addition, the core constructs of TTM (processes of change, decision balance, and self-efficacy) were applied to support progress in the changes across stages and facilitate behaviour modifications over the PHS-pro [32,36].

As far as we know, the PHS-pro is the first of its kind to apply the principles of TTM for developing skills towards healthy behaviours among Portuguese children.

### 1.2. Research Hypothesis and Objectives

Our research was focused on children aged 10–12 years, partly because this age group attends grade-6 classes and the Natural Sciences curriculum, and their school textbooks address topics related to healthy diet principles, the digestive system and its healthcare, digestion and the process of obtaining nutrients, which represents an excellent target for implementing nutrition-related education to promote the behaviour changes needed for a healthier lifestyle.

We have hypothesised that an educational programme focused on healthy eating and active living issues could encourage children to be active participants in their own change processes, improving their eating and lifestyle behaviours in addition to positively changing their body composition towards a better nutritional status. For the best evaluation of the PHS-pro, a mixed-methods approach was assumed, integrating quantitative and qualitative methods [37,38]. The PHS-pro evaluation included four central research objectives: (1) to assess the effects of the programme on the nutritional status of children; (2) to determine behavioural improvements (eating, physical activity) reached by children during the intervention; (3) to assess the costs and benefits of PHS-pro implementation; (4) to describe and evaluate the implementation process of the programme.

This study focuses on the fourth research objective, and the specific goals are to describe and evaluate: (i) the research design within the theoretical framework grounded on “The Transtheoretical Model” and the stages of change, (ii) the educational components and the application of the participatory methodology to engage children to meet their needs, as active participants in their change process, and (iii) the process evaluation of the intervention.

The first three objectives were accomplished with promising results. The PHS-pro did improve the anthropometric outcomes, effectively leading to better nutritional status, as well as effectively showing improvements in the eating behaviours of children, in particular in terms of the intake of fruit and vegetables. In addition, the cost–consequence analysis provided evidence that the PHS-pro was economically feasible, with an intervention cost of 36.14/year/child attending the programme, which is much lower than the medical costs for treating adult obesity. The effectiveness of the first three goals has been demonstrated in detail elsewhere [39,40,41].

## 2. Materials and Methods

### 2.1. Research Design

The research design was based on a pre-test and post-test structure with a non-equivalent control group; it is also referred to as nonrandomised control group pre-test–post-test design [42,43]. We employed a quasi-experimental configuration characterised by collecting measurements before and after the introduction of the experimental variable, which is, in this case, the implementation of the PHS-pro over a full academic year, to evaluate its effects compared to the non-equivalent control group. The word “non-equivalent” refers to non-equivalence due to the lack of randomisation [44]. The random selection of children from different classes and different schools allocated between intervention and control groups was not possible for the development of the research since the PHS-pro would generate cohesion and synergy among the participant subjects [42], so randomisation would create a bias caused by the daily contacts between subjects included in the group participating in the PHS-pro and subjects in the group not participating in it [44]. Thus, the framework of the research involved two groups (intervention and control groups), two assessments in the beginning of the school year (baseline) and at end of the school year (follow-up), and the educational intervention of the intervention group undertaken via the PHS-pro, as presented in Figure 1.

The overall research was undertaken in two main phases: Phase 1 involved the designing of the programme (educational components and measurement tools) and the overall preparation process with schools, over eight months, which allowed for the gathering of information about the local school children population, as well as the schools’ environments for the intervention; Phase 2, corresponding to the academic year of 10 months, involved the implementation of the educational programme and its evaluation.

### 2.2. Phase 1: Designing and Development of the Educational Components, Process Evaluation of the Programme and Preparation of the School Setting

The starting point of promoting healthy eating and active living in children was to set goals for the educational programme based on the international guidelines of the World Health Organization [45]: the adequate consumption of fruit and vegetables (F&V) in five servings per day; decreasing high-sugar food and beverage intake; decreasing high-fat and energy-dense food consumption; one hour of physical activity and reducing TV watching time below two hours per day.

Accordingly, to target these specific behaviour goals, and in order to produce a robust cognitive, attitudinal and behavioural impact, a list of topics was developed for designing the educational content of the eight learning modules and adapted to the TTM stages of change to be implemented during the intervention.

#### 2.2.1. List of Topics and Learning Modules Design

A list of 16 topics about food, eating and active life behaviours, and an extra free topic for an optional children’s suggestion, was prepared, following the participatory methodology principles [46]. Of this list, children would select eight topics (Appendix A Table A1).

The children’s participation was highly relevant to a successful implementation of the programme. Hence, children were asked which topics they were more interested in learning and developing over the programme to ensure their active participation and to meet their own needs, since they would be the major beneficiaries. The active role of children can help shape their own environment and enhance their skills, self-esteem and consciousness, while promoting a democratic and decision-making process [47]. In fact, the United Nations Convention on the Rights of the Child declared in Article 12 that the child “is capable of forming his or her own views freely in all matters affecting the child” [48].

As a result, the eight topics most voted on by the children were converted into eight learning modules, which constituted the core educational component of the PHS-pro.

The learning modules were designed to carry clear, concise and positive messages [49,50]. Instead of teaching nutrition facts or giving advice like “eat this, not that”, the central messages were based on children’s expressed needs to engage them effectively and communicate in a teen-friendly language to promote interactive discussions. Therefore, children could influence the PHS-pro content and the learning process with democratic and effective participation [51].

#### 2.2.2. Applying the Transtheoretical Model of Stages Change

TTM offers an understanding of how people develop new behaviours and improve skills and self-efficacy by assessing the stage of readiness for change and providing a set of processes of change or strategies to be applied in designing health interventions [32]. For this reason, TTM was set out as the theoretical support to design the PHS-pro and how the educational activities were applied in practice.

The core component of the intervention consisted of eight learning modules, which followed the five stages of TTM readiness for moving children from inaction to action or maintenance to promote a successful behavioural change. Therefore, based on the TTM constructs and processes of change, a conceptual framework was developed for each learning module to engage children to participate actively, to increase knowledge, and to develop competencies for decision-making related to healthy food and active life habits. Table 1 provides detailed descriptions of the TTM stages and their respective core constructs, the key strategies planned for the learning modules, and the expected process of change.

The process of behavioural change was assessed through the activities developed during the learning modules. To monitor the progress on the eating behaviour changes, a food record was applied after each module, which allowed the individual feedback of children [40].

Following the recommendations of Kristal and colleagues [52], the behaviour targets developed for the learning modules were a combination of general changes (e.g., avoiding fried foods) and specific changes (e.g., eating three fruits daily). Children’s stage of change was assessed to categorise the baseline stage in each class before the PHS-pro implementation. Children were asked whether they were seriously thinking about making changes in the next six months. The baseline stage of each child was categorised into four possibilities: (i) those with no plans to change any of the general behaviours presented—they were in precontemplation; (ii) those with plans to change without a commitment—they were in contemplation; (iii) those planning to change soon, in the next month, and set a behaviour goal—they were in preparation; (iv) children who reported trying to change and achieve the desired goal—they were in action/maintenance. These two stages (action and maintenance) were combined since the study had no time to assess children reporting the change for six months or longer. This procedure allowed the evaluation of the cognitive and behaviour engagement of children in the change process.

#### 2.2.3. Measurements and Process Evaluation

Measurements and process evaluation were conducted over one full school year, where multiple methods were used to determine the effects of the PHS-pro by comparing the results of the children intervention and control groups, as well as to capture the behavioural changes of children in the intervention group, from baseline (before the PHS-pro) to the follow-up assessments (at the end of the PHS-pro implementation).

To compare groups, data collection involved anthropometric measures, eating, physical activity and sedentary behaviours, and food knowledge. To measure change behaviours produced during the programme for the intervention group only, data were collected using: food records, feedback activities and questionnaires for children. Furthermore, questionnaires and interviews were also applied to teachers and children of the intervention group. Table 2 summarises the process evaluation methods, participants involved in data collection and types of measurements.

Measurements were carried out at schools at two evaluation times: baseline (during the two initial schooling weeks) and follow-up (during the two last schooling weeks). Each evaluation included: (i) an anthropometric assessment of the nutritional status; (ii) eating, physical activity and sedentary behaviours, and a food knowledge score assessed using a self-reporting questionnaire.

(i)Anthropometric assessment

Height, weight and waist circumference measures were collected during physical education lessons in sports clothes (shorts, t-shirt without footwear), using standard procedures [53]. Height was measured to the nearest 0.1 cm using a portable stadiometer (Seca model 214, Hamburg, Germany). Weight was recorded to the nearest 0.1 kg using a portable digital scale (Seca model robusta 813, Hamburg, Germany). Waist circumference was assessed in the anatomical landmark midpoint between the lowest rib and the iliac crest at the end of expiration using a flexible band (Seca). This measurement was taken three times, recorded to the nearest 0.1 cm, and we calculated the average of measures to improve precision [54,55]. To ensure measurement reliability, the equipment was calibrated prior to each day of assessments. All the anthropometrical measurements were obtained by one examiner (nutritionist and author of the research), and assisted by a teacher of the Health Promoter Office in recording the collected data.

Body Mass Index (BMI) values for normal weight, thinness, overweight and obesity were defined using cut-off points according to Cole and collaborators [56,57] to yield a complete picture of the prevalence of overweight and obesity in the studied population. Waist circumference (WC) was classified using UK reference percentiles to gather an overview of adiposity due to its correlation with intra-abdominal fat mass [58]. The waist-to-height ratio (WHtR) was calculated by dividing waist circumference by height in centimetres [55].

Raw BMI is considered a suitable method to assess adiposity change in children at risk of obesity [59]; nevertheless, height, weight and BMI for age and sex-specific z-scores were obtained using the LMS method [60] in cases of comparative analyses with international studies. BMI used alone to assess improvements in nutritional status is not reliable due to its low sensitivity to body fat percentage. In contrast, the WC and WHtR can more accurately identify fat tissue changes over time [61].

The anthropometric measures were used to demonstrate the effectiveness of the PHS-pro and served as evaluators of children’s nutritional status changes before and after the PHS-pro implementation, and for comparisons with the control group [39].

(ii)Eating, physical activity and sedentary behaviours, and food knowledge

A self-reporting questionnaire was used to collect the eating, physical activity and sedentary behaviours. Food knowledge score and socio-demographic variables were also obtained through this same questionnaire structured in five main parts: (a) socio-demographic variables included age, calculated on birth date related to each date of assessment (baseline and follow-up); education level and occupation of parents; household conditions; (b) food knowledge was assessed with one open question about the Portuguese food wheel and seven closed-ended multiple choice questions about the recommended food portions [62]. A score of 1 and 0 was assigned for each correct and incorrect answer, respectively, with a possible total score of 7; (c) physical activity included four items related to participation in out-of-school organised sports activities and the number of hours per week spent. Response options ranged from “never” to “7 or more hours” in a total of six categories; (d) sedentary behaviours were assessed with two items to ask the number of hours spent per day in watching TV and/or playing video games. Questions about physical activity and sedentary behaviours were adapted from the Youth Risk Behaviour Survey [63]; (e) eating habits were collected using a 58-item food frequency questionnaire (FFQ) based on a validated instrument for the Portuguese population [51].

This questionnaire was previously pilot-tested on an eligible sample of thirty children of the same region, grade and age, but who were not involved in the study.

Children were informed about the procedures and asked to fill out the questionnaire in 30 min at their Natural Science class under the supervision of the researcher.

(iii)Measuring eating behaviour changes

Another tool was selected to monitor the eating behaviours among children of the intervention group and evaluate the dietary evolution after each learning module and along the intervention. Thus, a 3-day food record was selected to collect the daily food intake and check the usual food consumption and choices in the intervention group. The application of this tool allowed children to report all the foods and beverages they consumed for three days after each learning module (two consecutive weekdays and one weekend day) in great detail: time and place, cooking method, quantity and brand, with a space to comment on anything else.

This allowed for measuring dietary change over the intervention and according to the multiple target behaviours of the PHS-pro. The food record-obtained information should answer the following four questions: (i) Did intervention children improve their consumption of F&V? (ii) What beverages mainly contributed to children’s daily diet? (iii) Did energy-dense food consumption (high-fat and high-sugar food) and (iv) soft drink intake decrease over the intervention?

Each food record determined the effectiveness of each module in the short and long term. These results on the effectiveness of each educational module, using the food record tool, are published elsewhere [40].

(iv)Process evaluation

The evaluation of the intervention process was conducted by assessing the insights of the participants about the programme in terms of its educational components and dimensions. Two open-ended questionnaires were designed, one for children and the other for teaching staff. Additionally, at the end of the PHS-pro, a Strengths, Weaknesses, Opportunities and Threats analysis (SWOT) was conducted with the teachers who were directly involved in the programme’s implementation. This feedback allowed for an understanding of the perceptions of the participants and allowed us to identify key facilitators, as well as the constraints and limitations of the PHS-pro’s implementation.

The questionnaire given to children was intended to collect their views about the implementation, and it was applied shortly after the end of the last learning module of the programme. First, children were asked to give their opinions concerning the components of the PHS-pro, focusing on what they liked less, the reasons, and suggestions for future programme improvements. Second, children were specifically asked whether they recognised any changes in their own eating behaviours over the intervention by a yes–no question, followed by an open space to report what changes, if any.

The open questionnaire for teachers allowed for documenting how they perceived the PHS-pro intervention applied in their school. The questionnaire asked them to give feedback about the different components of the programme, giving particular attention to four dimensions: the PHS-pro’s organisation, learning modules, food records and literary contest. In addition, teachers were asked to suggest improvements to the programme. The questionnaire was applied to all teachers of grade-6 involved in the PHS-pro.

The SWOT analysis was also given to teachers to derive an effective overview of the components, dimensions and strategies adopted during the implementation, regarding their involvement in the PHS-pro’s activities as well as the delivery of the programme to children.

A focus group interview was also conducted with the involved teachers to understand their experiences with the PHS-pro and their perspectives. Data were structured according to the four SWOT categories: strengths, weaknesses, opportunities, and threats. The diagnostic power of SWOT would enable the identification of gaps and best practices, and provide useful inputs to improve the planning process of the PHS-pro for future interventions [64,65].

#### 2.2.4. Preparation of School Setting

For carrying out the research, the preparation process of the school setting involved different procedures and stakeholders: (i) recruitment of schools; (ii) ethical aspects; (iii) school setting conditions; (iv) sessions with parents and teachers.

(i)Recruitment of schools

All four elementary schools in the urban municipality (Trofa), integrated into the second largest metropolitan region of Portugal (Porto), were invited to participate in the research, as our first aim was to cover all children attending grade 6 in the municipality.

In order to organise two balanced samples of subjects, an estimation of the number of children usually registered was calculated for the next school year to implement the PHS-pro and arrange a reasonable sample with two homogeneous groups: the intervention and control groups. One of the four schools had registered more children, and consequently more classes, with whom to implement the programme. For this reason, it was selected as the “intervention school”, whereat children in grade 6 composed the sample of the “intervention group” (IG). The three remaining schools were selected as “control schools”, whereat the children in grade 6 comprised the sample of the “control group” (IG). Being a quasi-experimental study, the PHS-pro was applied to children of the IG only, at the intervention school; the CG did not participate in the PHS-pro and children therein were only subjected to the two evaluations (baseline and follow-up assessments).

(ii)Ethical aspects

The research followed the ethical standards recognised by the Declaration of Helsinki. The Scientific Council of the Institute of Education of the University of Minho approved the PHS-pro, and ethical permission was obtained from the Pedagogical Board of each school.

School principals and class coordinators of all schools agreed to collaborate in this research project. Also, a partnership was established with the Health Promoter Office (HPO) of the intervention school, enabling the inclusion of the PHS-pro in their “Yearly School Activity Plan”, which facilitated synergies to develop other activities in the school, not initially planned. As a result, the HPO coordinator made all the arrangements required for implementing the intervention in the school setting, connecting the PHS-pro with several stakeholders: class coordinator teachers, grade 6 teachers, the school principal, the school parents’ association and the school library. 

The PHS-pro schedule was delivered at the four schools, covering all assessments and activities required to conduct the research and allowing integration between the coordinator teachers and class teachers for the regular running of the school schedule. The advanced planning allowed the researchers to be present at all crucial moments for granting authorisations and providing full information concerning the research. 

Parents were invited to participate in the school meeting and were fully informed about the study procedures, participants’ risks and benefits, access to information on the child’s nutritional status at the end of the programme, and data confidentiality. Parental written informed consent was obtained for the participation of their children in the PHS-pro. 

A similar structure was delivered to children in the classroom on the first day of the school year. They were informed about the respect for privacy and the confidentiality of the data collected during the research process. A detailed explanation was given first verbally, then through the distribution of a paper package that included a participant information leaflet about the research purpose and an informed consent form for the child to enable free and active participation beyond the formal request previously addressed to parents. Thus, a range of ethical precautions were taken into account and ensured by the ethical principles of the participatory methodology [66].

(iii)School setting conditions

Several factors that might affect the research outcomes were controlled in the four schools: school grade, nutrition-and health-related school curriculum and textbooks, physical education classes, and school food services. It ensured homogenous conditions among children from the different schools. Thus, the food infrastructures of the involved schools, such as vending machines, cafeteria, and the food menus, were analysed. All four schools exhibited identical conditions: children had access to the buffet during breaks with a similar food offering; cafeterias presented lunchtime menus, which followed the guidelines of the Portuguese Ministry of Education [67,68]. However, there was a difference in the management of the cafeterias: the three schools comprising the control group were run by catering businesses, whereas the intervention school had internal management, which allowed, over the PHS-pro implementation, the support of the school direction to guarantee the availability of F&V in the school canteen at lunch, with an extra salad buffet.

Curriculum programmes were assessed, specifically the grade 6 Natural Sciences curriculum and its textbooks, to understand whether children would be subjected to identical conditions over the school year. Textbooks, didactic content and time structure were followed identically under the guidelines published by the Curricular Organization and Programs of the Ministry of Education [69]. Additionally, subjects related to the digestive system, nutrition and associated areas were taught by the four schools at the same time during the first school period, avoiding methodological biases between the intervention and control groups.

(iv)Sessions with parents, teachers, and other school staff

The briefing session organised for parents also marked the commencement of the PHS-pro. After ethical procedures, parental involvement was requested to raise awareness of the importance of healthy eating at home. Educational messages delivered to parents to take home were based on: (i) eating fruits and vegetables every day; (ii) eating vegetable soup to start lunch and dinner; (iii) increasing fish intake and decreasing meat; (iv) healthy food choices available between meals; (v) drinking primarily water; (vi) having breakfast; (vii) being active at least one hour a day.

Parents were challenged to adopt these healthy behaviours, leading the example and contributing to the healthy growth of their children. In addition, this session was intended to promote equal attitudes in the family environment over the intervention period. Parental attendance reached a rate of 100% at this session. The parents and children were unaware about the existence of intervention and control schools, and consequently, of the two studied groups of participants.

The briefing session for the parents in the intervention group had an additional goal: to encourage parents to increase F&V availability at home and support children in making behavioural changes while implementing the PHS-pro. Expressly, the key message strongly advised having available at home: fresh fruit; vegetable soup at meals, mainly at dinnertime; and salads or vegetables as side dishes.

The teaching staff of the intervention school was invited to the meetings of the PHS-pro to request active participation, listen to all the point of views, and invite teachers to participate in the challenges of the PHS-pro. Such challenges related to the fact that the PHS-pro would run over the academic year and could organise, in partnership, school activities related to eating and healthy lifestyle behaviours. Teachers responsible for Natural Science, Portuguese Language and Physical Education worked more closely in the PHS-pro’s implementation, and were informed that different tasks would be required of them.

Furthermore, one meeting was set up for the kitchen staff of the school canteen (a cook and two auxiliary staff), where the purpose of the PHS-pro was explained and collaboration was requested to ensure at the lunch meal the availability of fruit, vegetables, and salads ready to eat.

#### 2.2.5. Studied Population and Sample Selection

There were two main reasons for selecting children attending grade 6 classes for the development of the educational programme. The first is because the grade-6 Natural Science curriculum and school textbooks address issues related to nutrition and nutrients, healthy diet principles, the digestive system, digestion and the process of obtaining nutrients, and healthcare for the digestive system. Thus, both intervention and control groups were ranked at the same knowledge level at school, and were also provided with the appropriate environment to expand their awareness of several of these issues, encouraging children to be aware of eating matters and simultaneously increasing children’s ability to choose healthy food and to know more about what they really eat [70]. Secondly, this age group (mainly 10 to 12 years old) shows greater autonomy, including financially, compared to the younger ones who attend the earlier elementary school years. In grade 6, Portuguese children are immersed in an environment without parents’ supervision on snacks, and have access to the school buffet, a new reality that increases the risk of unhealthy food choices in the school and outside the school. Therefore, this research proposed to recruit all children who attended grade 6 in the elementary schools of one Portuguese urban municipality.

Table 3 presents the criteria for the children participants’ inclusion or exclusion.

A total of 504 boys and girls (aged 10–14 years old) were found to be registered in grade 6 at the four schools of the studied municipality and were enrolled in the study. All parents signed the informed consent, and all children agreed to participate freely, except one boy from the intervention group school, who refused to participate in the research.

Of these 504 children, 55 (10.9%) were excluded: 1 refused to participate, 5 moved to another municipality’s school, 21 had special education needs, and 28 did not attend both assessments (baseline and follow-up). Children with special educational needs participated in all tasks and educational activities carried out by both groups (CG and IG), but their data are not included in the analysis. Therefore, the final sample comprised 449 children, 89.1% of the available population, with 219 children (48.8%) in the intervention group and 230 (51.2%) in the control group, as shown in Figure 2. In short, the control group was limited to the regular teaching of the Natural Sciences curriculum by studying nutrition and related issues in the classroom, and was submitted to both assessments for data collection. In contrast, the intervention group participated in the PHS-pro over the school year, and was also submitted to both assessments.

### 2.3. Phase 2: The PHS-pro Implementation

The PHS-pro intervention started in the initial schooling week and continued until the last. The intervention school (intervention group) included nine classes of grade 6 that participated in three learning activities: (i) learning modules; (ii) the workshop “Cooking is Science”; (iii) a challenge—literary contest. The PHS-pro was initiated with a briefing session and monitored via seven food records (Figure 3).

#### 2.3.1. Briefing Session for Children

The starting point of the PHS-pro intervention was a briefing session (BS) in each grade 6 class. The different components of the programme, schedules and rules for effective functioning were introduced to children.

Since the PHS-pro sought to create a time and a space for learning healthy behaviours via a practical approach, it was emphasised that the intervention would be subjected to children’s acceptance and opinions, according to the procedures of participatory methodology [33,38]. Therefore, the approach to the learning modules’ implementation was explained, and children were told they would have time to share their thoughts and doubts and explore their expectations.

During the baseline assessments and after the application of the self-reported questionnaire, children chose the eight topics they wanted to study in the learning modules, and this briefing session was also used to informed children about the eight-topic list. In addition, the structure of the 3-day food record was explained to children, and how to fill it in correctly and the delivery rules were defined. The first food record was delivered and applied in the following days. This first food record is designated as food record zero (FR0), and allows the identification of foods and beverages consumed and the monitoring of the usual eating patterns of children before the intervention. The FR0 represents the basis for comparing the subsequent food records over the intervention. Indeed, after each learning module, a food record was distributed to children to be filled out within the subsequent three days (see Figure 3) and returned to the class teacher at the next lesson. This procedure was repeated till the end of the intervention except in the last module, where a game (G) closed the activity instead.

#### 2.3.2. Baseline Stage of Change

The baseline stage of readiness to change behaviours was assessed in children. The target was to evaluate their cognitive and behavioural engagement before initiating the first learning module of the PHS-pro. For this, general healthy habits were structured in a list entitled the “10 steps to be healthier”: (1) daily breakfast intake, (2) eating vegetable soup to start lunch and dinner, (3) eating a colourful diet, (4) eating more fruit—three different fruits each day, (5) balancing meals between meat and fish, (6) eating two snacks midmorning and afternoon, (7) avoiding high-fat food, (8) reducing salt, (9) drinking water, and (10) regular physical activities and watching their weight.

Based on the questions and algorithms derived from Kristal et al. [52] in relation to stages of change, the baseline stages of children for the “10 steps to be healthier” were categorised and assigned according to the five stages of TTM: (i) precontemplation (not seriously thinking about changing habits), (ii) contemplation (seriously thinking about changing habits, still not sure when), (iii) preparation (planning to change soon, in the next month), and (iv/v) action/maintenance (currently trying to change).

The “10 steps to be healthier” were introduced in each class of the intervention group, and children were first asked about their intention to adopt those healthful habits (Figure 4).

#### 2.3.3. Implementation Process of the Learning Modules

As mentioned above, children selected eight topics they wished to learn about during the intervention. The contents of the eight learning modules as defined by that selection were developed and organised into this suitable sequence to ensure appropriate action and allow the support of the TTM model’s assumptions.

The eight learning modules were the following: (1) “10 steps to be healthier”; (2) “Water & milk help you to grow up”; (3) “Training every day to be healthier”; (4) “3 fruits a day, how much good it does?”; (5) “F&V are essential to life”; (6) “Start on moving!”; (7) “The best snacks”; (8) “Final game: who has learned about everything?”

These learning modules were designed to improve food knowledge, food choices, eating behaviours and other healthy habits, as well as to develop skills in children, and help them think about their real problems and learn how to deal with them. 

Each module lasted approximately 30 min and was run monthly in the Natural Science classes without interfering with school activities. The eight modules, previously scheduled for the first hour of a two-hour class, were established with each teacher.

The general structure of the learning modules included a short presentation of the topic and the behavioural goal, time for discussing and sharing children’s experiences, feelings and opinions, an activity related to the topic, and a behaviour goal setting proposed by children to be applied for the coming learning module. The implementation process of an exemplar learning module is illustrated in Figure 5. 

After the first module, each consequent learning module was initiated with feedback from the previous module regarding children’s improvements and commitment goals.

Further details on the contents of the eight learning modules, the activities undertaken, and the outcomes to be achieved during the programme are already described elsewhere [40].

Educational materials included PowerPoint presentations, worksheets and paper cards to draw with during the activities developed in the learning modules. 

The eight modules were conducted by the researcher (nutritionist) and supported by the teacher of Natural Sciences. In addition to being an observer and having a moderating influence, the teacher also coordinated the time management in the classroom so that the researcher adopted the facilitator role in conducting the short discussion needed. Additionally, in every module implemented in each of the nine classes at the intervention school, data were collected and gathered in a fieldwork notebook, allowing for registering children’s perceptions and comments in a real-time class. 

The monitoring of the learning modules was conducted regularly via the application of food records (see Figure 3). This procedure was repeated over the intervention with seven food records, allowing for monitoring the outcomes of the corresponding module.

#### 2.3.4. Workshop “Cooking Is Science”

The workshop “Cooking is Science” was an activity that arose from the partnership established with the intervention school’s Health Promoter Office and allowed for the creation of synergies among teachers of grade 6 classes.

Teachers were invited to organise activities related to healthy eating and lifestyle habits during the PHS-pro intervention period. Following this, the teaching staff organised a set of 2-day workshops in the second period that were open to all school grades, in which topics related to food were linked to different science areas in an interdisciplinary approach, under the title “Cooking is Science”, and with six topics: “everything in proper measure”; “fruits are health”; “how is bread produced?”; “there are delicacies that are bacteria’s work”; “fresh cheeses”; “how did our grannies preserve food… without a fridge?”.

The PHS-pro selected the topic of “fruits are health” and was responsible for developing a culinary activity with strawberries, one of April’s seasonal fruits. 

The activity started with a few recommendations: introducing this fruit at breakfast and explaining the need for a generous amount of fresh strawberries to benefit from their great nutritional value. Then, a simple strawberry recipe was cooked to be consumed as a snack. Under the supervision of the school staff, the children prepared the strawberries, followed the recipe and tasted it. This activity aimed to provide contact with strawberry preparation and develop the ability to wash and cut this kind of fruit. It created opportunities to understand children’s perceptions about strawberries, as regards preferences, potential uses, nutritional quality, and food safety.

#### 2.3.5. Challenge: The Literary Contest

Children were challenged to participate in a literary contest (LC) to continuously follow up on the main focus of the PHS-pro over the year: adopting healthy behaviours. This activity was intended to encourage self-expression and creativity in writing and/or drawing, providing opportunities for children to express their ideas, opinions, day-to-day experiences and emotions about healthy eating and active living issues without forgetting to give voice and agency to participants [71]. Therefore, a story-reading session about eating and lifestyle behaviours was undertaken during the Portuguese language class. Then, children were invited to participate voluntarily in the LC: to write a short story in the form of a comic strip, with illustrations or other artistic features, where they could develop and work on an issue related to healthy habits in group or individually.

This activity was developed with the collaboration of the Portuguese language and Visual and Technological Education teachers and coordinated by them over nine months (see Figure 3). The rules of the LC were presented, giving children six months to build the storyline and submit their creative writing to the jury committee. Short stories that met the requirements were selected and evaluated by a jury composed of eight elements (two fiction writers, a children’s literature lecturer of the University of Minho, a school teacher and four school students of grades 5, 7, 8 and 9).

The LC culminated at the closing event held at the end of the school year to exhibit all the works and announce the winning short story.

#### 2.3.6. Data Analysis

Data collected over the PHS-pro research were organised around the four central research objectives (see Section 1.2). As defined above in a mixed-method evaluation process, data were derived from different sources—anthropometric measures, questionnaires, food records, semi-structured individual interviews, teachers’ focus group, field notes, and documents about the schools’ contexts.

The method of data analysis for the first three research objectives has been described in detail previously [39,40,41]. In this paper, whose objective is to evaluate the implementation of the PHS-pro from the point of view of the participants, the analysis focuses on the qualitative data gathered from the participating children and teachers, which information was entered into a database in Microsoft Excel. Answers to the open-ended questions were examined according to the components of the PHS-pro, and categorised using a content analysis methodology [72]. Field notes, information, and documents gathered over the research served as the basis for the analysis, and results are organised according to the study Phases 1 and 2.

The results and findings of this behavioural change programme adhere to the CONSORT guidelines and the checklist for reporting non-pharmacological interventions [73].

## 3. Results

### 3.1. The PHS-pro Implementation and Findings

#### 3.1.1. General Characteristics of the IG Participants

The PHS-pro sample (the intervention group) was composed of 229 children, 119 boys (52%) and 110 girls (48%), with a mean age of 11.0 (SD = 0.7) years. Children were distributed in the nine classes of grade 6, with a minimum of 20 and a maximum of 28 per class.

Sixteen teachers were permanently involved in the PHS-pro school intervention: five teachers of Natural Sciences, four of Physical Education, six of Portuguese language and one of Visual and Technological Education, and the coordinator teacher of HPO was also included.

#### 3.1.2. The Eight Topics Selected by the IG Participants

Children could select 8 topics out of the 16 suggested and 1 free topic they could propose. Of the 229 children, 100 (43.7%) chose one free topic each, which were collected into three categories: (i) healthy eating and specific food questions (59 children; 59%), (ii) sports and physical activities (32 children; 32%), and (iii) dieting and weight loss (9 children; 9%).

Thus, the eight topics that were most voted for to be used for learning modules are shown in Figure 6. The less voted-for topics were those related to sugar and sweets, suggesting that children are more interested in developing healthy competencies and learning healthy choices than those already known as unhealthy.

#### 3.1.3. Baseline Stage of Change of IG Participants

The intention to change behaviours in children was assessed at baseline in each of the nine classes of the intervention group. According to the “10 steps to be healthier” list (see Section 2.3.2), none of the children were indicated to be in the precontemplation stage, as shown in Table 4. In all classes, at baseline, children reported trying to change three of the ten steps (1; 6; 9) considered for these behaviours in the action/maintenance stage. Children in all classes were in the contemplation stage, and were thinking about changing their habits in steps 7 and 8. Also, children of all classes, except one (class F), were in the preparation stage for steps 2 and 10, as they were planning to change soon. For the other steps (3, 4 and 5), children ranged between the two stages: contemplation and preparation (Table 4).

These findings show that children were thinking about changing or were prepared to change some of their eating behaviours among the ten steps presented, which was promising for their involvement in the PHS-pro.

#### 3.1.4. The Food Records Application

After the seven types of 3-day food records were given to the sample of 229 children, 1089 food records were collected, corresponding to 3267 days of food reports. The summary of the collected data is given in the following:-70 children (30.6%) returned all seven of their food records completely filled in, with the greater involvement of girls (42 = 60.0%) as compared to boys (28 = 40.0%). Only 12 children (5.2%) did not return any food record;-Most children (72.1%) returned four or more food records, and 27.9% returned fewer than four. Again, girls had higher rates of delivering food records (54.0%) than boys (46.0%), as seen in Table 5.

Figure 7 shows the return rates of the seven food records after each learning module. The return was consistently above 60%, except for food record number 6 (FR6), which was returned just before the Easter school holidays, and declined to 44%.

#### 3.1.5. Literary Contest (LC)

Of the 227 children, 165 (72.7%) enrolled voluntarily to participate in the LC, as a group or individually, for developing short stories. Sixteen stories met the rules of the contest and were accepted for jury evaluation. This activity involved 41 children from the nine classes.

The winning story was written by four boys belonging to Class H, and it was shared with all classes and grades to celebrate the end of the school year.

#### 3.1.6. Workshop “Cooking Is Science”

Children of all nine classes participated in the workshop “Cooking is Science” (*n* = 227). This activity involved 28 groups of eight children each. All children agreed that strawberries are one of their favourite fruits; however, they never took part in hands-on recipe preparation with any kind of fruit, especially those to be consumed as a snack. In addition, all groups of eight children showed difficulties in dealing with the washing and the cutting of strawberries. During the tasting of the recipe, the majority of children expressed interest in reproducing the recipe at home to share with their family.

### 3.2. The PHS-pro Evaluation by Children and Teachers

#### 3.2.1. Children’s Views on the PHS-pro Intervention

The participation rate of children in the PHS-pro school intervention was 99.1%, i.e., of the 229 children, 227 completed the questionnaire.

The food record application was the PHS-pro component that children liked the least (Table 6). Of the 227 children, 116 (51.1%) said that food records are “too much work to do”, and also complained of “too many days to record”, “whenever I eat, I have to record”, or “I do not like to write”. On the other hand, three children (1.3%) reported that food records helped them “to pay attention to what they were eating”. By contrast, among the 227 children, 82 (36.1%) did not pick any of the boxes available, stating they liked to participate in everything.

Although children considered the 3-day food record a tedious task, the recovery reached high rates during the relatively long intervention period (see Figure 7). As a group, girls were more engaged than boys in the recording process, delivering more information about food and beverage consumption (see Table 5).

The perceptions of children about their behavioural change showed that most (84.6%) recognised changes in some eating habits (see Table 6). They mentioned that they have started eating more fruit (40.6%), followed by vegetable soup (27.6%). In fact, these most frequently declared behaviour changes match the main focus of the learning modules developed over the PHS-pro intervention regarding improving fruit and vegetable consumption.

Table 6 also shows that children were pleased with their participation in the programme, as most of them (98.2%) agreed that this kind of programme could help other children adopt healthy behaviours, with nearly half of the children (45.0%) indicating that the PHS-pro is appropriated for helping young people to eat right.

These findings suggest that children should be encouraged to participate to increase their motivation, and their inputs can lead to a better acceptance of the intervention and better outcomes.

To support the importance of children’s motivation to the implementation process, an informal conversation documented in the researcher’s field notes is reproduced below. The conversation took place at the school’s entrance in the mid-term of the PHS-pro intervention (see Figure 3) between the researcher and three children participants:

Children 1: Is it today you’ll come to our class?

Researcher: No, it is on next Wednesday.

Children 1: Oh, it ought to be today!

Researcher: So, it’s only a couple of days away.

Children 1: But it should be today.

Children 2: Don’t you see it is in the two-hour class? It is better. Yeah!

Children 1: For over a month now, I don’t eat French-fries!

Children 3: So don’t I.

Children 2: Me too, almost for a while. At least I eat less.

Researcher: Fine. So your class are already having healthier changes.

#### 3.2.2. Teachers’ Evaluation of the PHS-pro Intervention

Teachers participating in the implementation were asked to contribute their perspectives and opinions about the programme’s components in which they had collaborated directly. Of the 16 teachers, 12 (75%) answered the questionnaire and participated in the focus group for a SWOT analysis. Table 7 shows the 12 teachers’ remarks on the PHS-pro components. Teachers mentioned more often the literary contest as a good and successful experience involving the children (with nine inputs). In contrast, the seven 3-day food records were considered a challenging activity at first, but became tedious, an opinion similar to the children’s opinion (see Table 6).

The learning modules and their organisation were, in general, well accepted (Table 7).

The answers to the questionnaire applied to teachers were very useful for initiating the discussion with them during the focus group for the SWOT analysis. Table 8 summarises the strengths, weaknesses, opportunities and threats identified in the overall PHS-pro intervention. Generally, teachers reported the implementation as a positive experience. All teachers mentioned that it is crucial to have this kind of intervention based on learning and changing to develop healthy behaviours. They suggested additional components (more interaction with parents, more activities, more collaboration of school staff, and more coordination), which will demand increased coordination. In addition, they agreed that the programme should be continued, and extending to other grades. However, commitment and motivation to collaborate over the intervention varied among the teacher participants.

## 4. Discussion

The current study aimed to describe the step-by-step development of the PHS-pro, a behaviour change programme, in which the research design, the theoretical support of the educational components implementation, and the process evaluation were addressed. Educational components and related assessment tools were designed and selected to achieve the following goals: improve fruit and vegetable consumption by making children aware of the importance of an adequate intake of such foods and guiding them to other healthy choices each day.

The PHS-pro was designed under the TTM of behaviour change, which has been applied to multiple health-related behaviours and demonstrated to be an effective and flexible model for promoting behaviour changes, particularly in healthy eating, physical activity, and fruit and vegetable intake [31,74]. The TTM of behaviour change guided the process of change, allowing for interpreting children’s stage of change, such as how children were specifically prepared for changing [33]. In fact, to improve the likelihood of effectiveness, interventions need to be based on behaviour change theories that should focus on how to promote action [75]. In addition, the baseline stage of change of children allowed for clarifying the stage of readiness to change behaviours and conduct the PHS-pro according to their intention of changing (see Table 4). This strategy unlocked an important answer concerning children’s real needs, which was consequently critical to the development of PHS-pro contents for developing knowledge and skills in children and facilitating the adoption of adequate eating and healthy behaviours.

Another successful approach of the PHS-pro conception was the application of the participatory methodology principles [46,51], seeking the active participation of children in their own change process. Active participation and the understanding of needs and interests were a permanent concern of the PHS-pro, so that children could build and develop skills for adopting adequate eating behaviours. Indeed, children were invited to take part in the programme’s activities by selecting the topics and participating actively in the learning modules by expressing attitudes, opinions, and spontaneous doubts. Moreover, the challenge of creating short stories in a literary contest and as part of the workshop “Cooking is Science” were included and conceived to be pedagogical helpers, providing participation, promoting cohesion and increasing self-determination [66,76].

To evaluate the programme as a whole, several strategies were combined to collect different data from children and teachers. The points of view provided by children about the programme’s components and their perceptions of behavioural changes showed a good acceptance and compliance, with the entire sample agreeing with the overall educational components: learning modules, literary contest and workshop participation. The selection of topics, learning module participation and the related activities were generally agreed upon by the entire sample, and were deemed successful among children because the topics were adapted to their profile and their perceived needs, which also simplifies the process of motivating behaviour changes, as acknowledged in the assessment of the baseline stage change. The literary contest was accepted by 73% of children, despite only 18% (*n* = 41) being submitted to the final contest. By contrast, the 3-day food records were the least approved activity; however, the rate of returning the food records was consistently above 60% during the intervention.

Additionally, the perceptions of children and teachers regarding how the programme’s components were conducted and reached them during the intervention were examined. Of all the components, the food record application was considered tedious and received the worst opinion from a large part of children, as confirmed by the teachers. This reaction could be justified because the food record was applied several times, becoming repetitive over the intervention. This repetitive task was considered by the teachers as one of the threats to the PHS-pro’s implementation. Although monitoring with this food record tool was crucial to evaluating the PHS-pro’s effectiveness, this activity was also recognised as a specific barrier to sustainable implementation in the long term. Indeed, the food record was an assessment tool that allowed access to children’s food choices and eating patterns, which helped us to identify trends, draw conclusions and provide information for future interventions [40]. Such findings can assist in improving the process of collecting information continuously for monitoring behavioural change, showing that other potential strategies must be investigated in place of the food records to be integrated into future interventions.

Children’s perceptions about their own eating behaviours during the implementation programme showed that they could identify positive changes, indicating improvements in fruit and vegetable soup consumption. It can explain why most children, when asked to give their opinion about the intervention, supported the idea that the PHS-pro can help young people eat healthier.

Teachers highlighted several strengths and opportunities of the programme, confirming the successful strategies implemented, and reinforced by the suggestion of extending the programme to other school grades. Also, threats and weaknesses were identified, in particular the need to reach closer to parents during the PHS-pro intervention, and highlighted the influence of the teachers’ motivation on the PHS-pro, which can strengthen the intervention or might become a great barrier whether there were difficulties in reaching to some teachers and motivate them to participate. These findings are very useful in developing improvements to be incorporated in future applications of the PHS-pro to increase participants’ acceptance and be sustainable in long-term interventions. Besides, these practical suggestions provided by participants should be considered in designing future educational programmes.

Another strength of the PHS-pro was the early planning of the programme. Conciliating the schedules of all participants and stakeholders ensured that all components of the programme could be considered during the conceptual planning and embedded in the school year’s activities. Another strength is related to the research design, which was able to gather two similar groups of children and allowed us to evaluate the effectiveness of the PHS-pro in the intervention group school compared to the control group schools, with results already published elsewhere [39].

The self-reported data of children gathered in the questionnaires are the main limitation of the PHS-pro approach, as they might leave room for subjective interpretation. However, to minimise this, an inclusive process of data collection was chosen, and categories were created (see Table 6) to reflect children’s inputs of each evaluated component [72] with additional triangulation with the points of view of the teachers [77] (as presented in Table 7). A second limitation was the use of qualitative questionnaires that were not previously validated. Since the programme was implemented for the first time here, the questionnaires were tailored to analyse the components of the programme and provide an exploratory approach to the testing of the preliminary intervention. More committed participants may have selected more positive answers; however, these questionnaires sought inputs on what went wrong during the implementation, searching for barriers and limitations to improve the programme rather than pointing out what worked well. Therefore, it was possible to decrease the risk of bias by requesting negative observations from participants [78]. Third, although parental involvement was included, it was restricted to two moments, at the PHS-pro’s beginning and its end. This strategy was partly intentional, because this PHS-pro was a pilot version, and it is of the utmost importance to investigate the impacts of the components and strategies designed to improve children’s skills in making healthy choices. Furthermore, the PHS-pro sought to help children to develop their responsibility in real-world environments, rather than increasing the parental supervision of issues that children must learn and deal with. In future interventions, the PHS-pro should include strategies of communication and educational tools to improve parental skills, which can help them in supporting their children. Also, teachers and the school community can participate more broadly, although there will always be some who will not be receptive.

## 5. Conclusions

The current study indicated that the PHS-pro methodological approach, planned according to the TTM and participatory methodology, can be used to effectively develop healthy eating behaviours and guide young people to healthy growth. The participation of children must be closely considered in order to increase their motivation, and their early input can lead to a better acceptance of the intervention, and better outcomes. Also, this study contributes to the body of knowledge in the largely unexplored area of interventions for preventing childhood obesity, and therefore offers evidence to guide the practice of health promotion in schools.

To our knowledge, the PHS-pro is the only formal Portuguese programme model to apply the stages of change and the core constructs of the TTM in developing skills related to healthy behaviours among Portuguese children over a full school year. This research is important from the health promotion perspective, because it examines new, dynamic approaches to implementation in the school environment, and has opened new pathways of action for preventing overweight and obesity among children.

## Figures and Tables

**Figure 1 nutrients-15-04543-f001:**
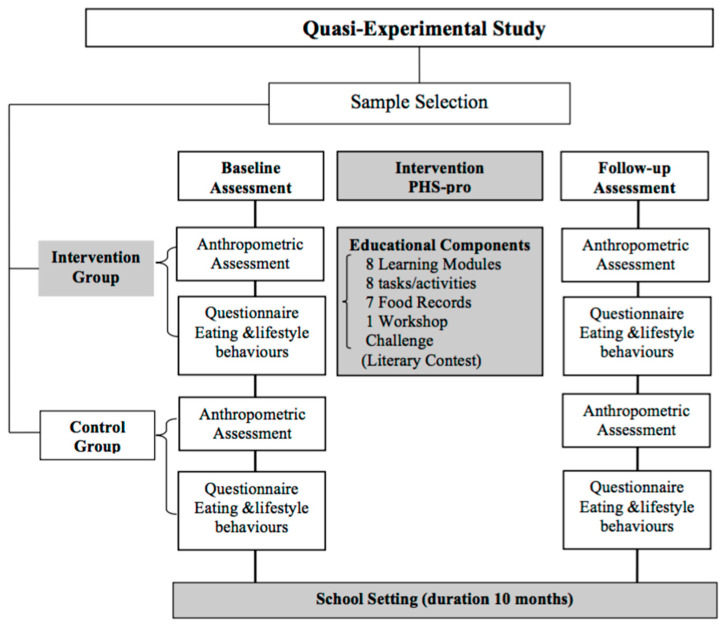
Flow chart of the study design.

**Figure 2 nutrients-15-04543-f002:**
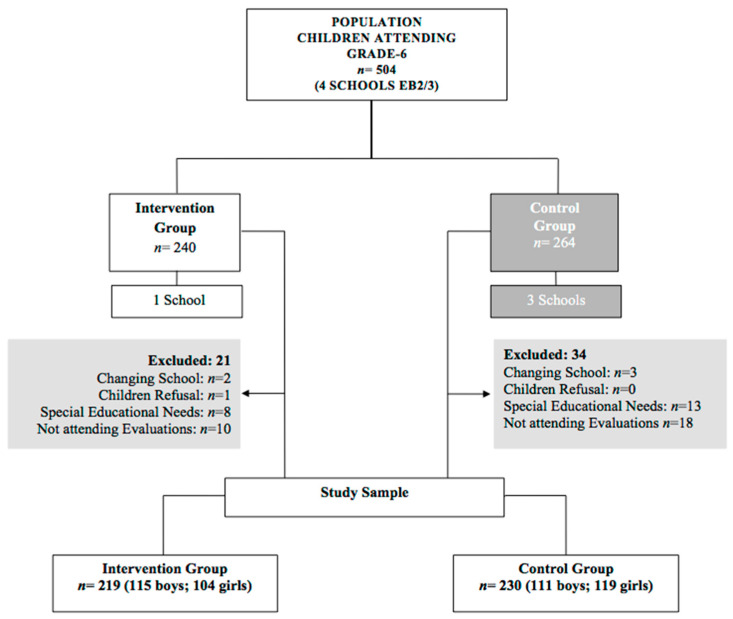
Flow chart of the sample selection.

**Figure 3 nutrients-15-04543-f003:**
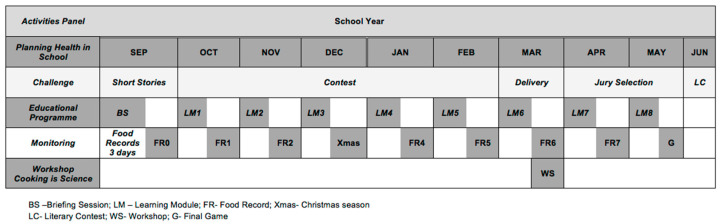
Intervention timeline of PHS-pro.

**Figure 4 nutrients-15-04543-f004:**
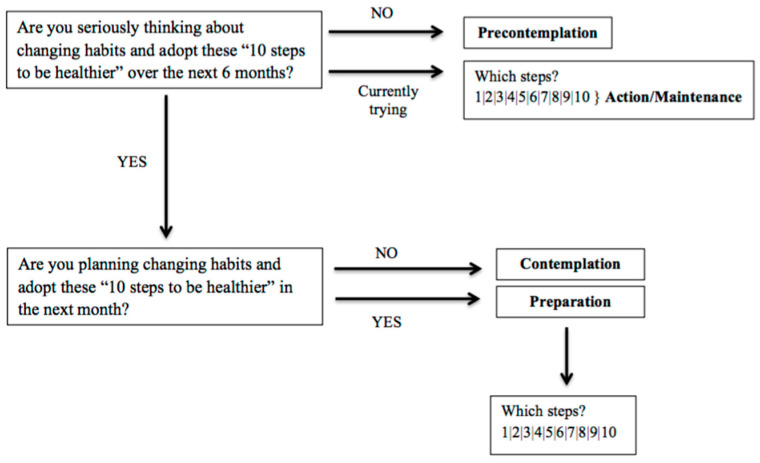
Questions used to assign stages of change.

**Figure 5 nutrients-15-04543-f005:**
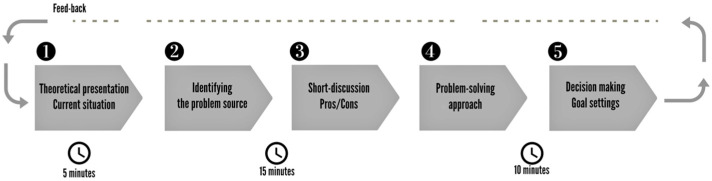
Implementation process of a learning module.

**Figure 6 nutrients-15-04543-f006:**
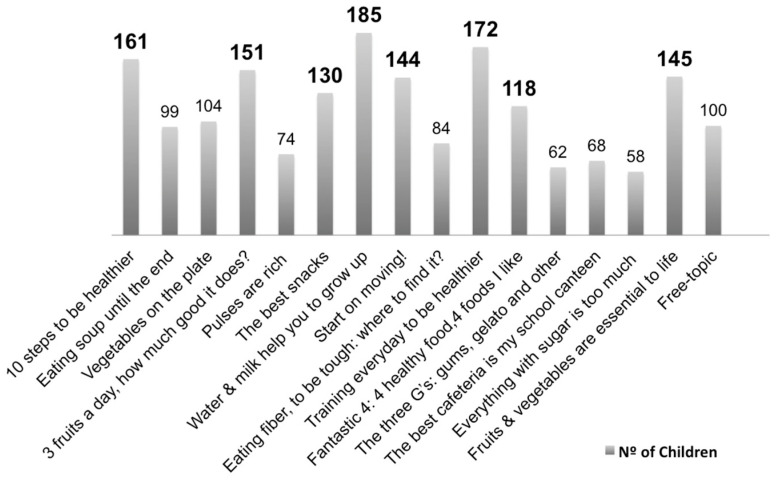
The eight topics (in dark) selected by children.

**Figure 7 nutrients-15-04543-f007:**
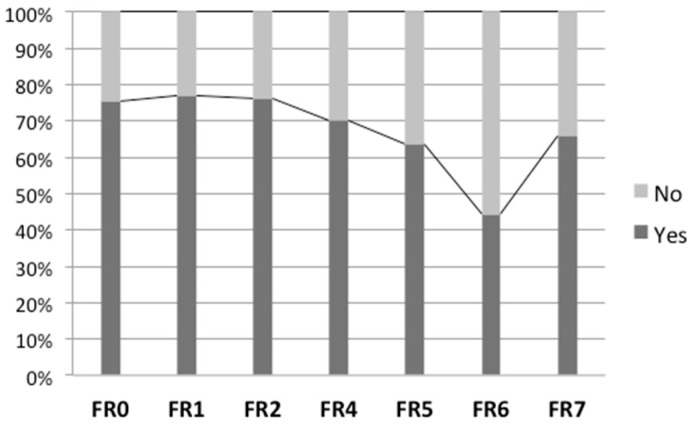
Food records return percentages.

**Table 1 nutrients-15-04543-t001:** Stages and processes of change applied in the eight learning modules, adapted from Prochaska, Redding [33] and Prochaska, Norcross [32].

Stage of Readiness	Description of the TTM’s Core Constructs	Key Strategies for Learning Modules	Process of Change
Stage 1 Precontemplation	No intention to changing behaviour in the near future (in the next six months). Unawareness of their problems. No interest in changing behaviours. No plans to change.	Learning module 1 - Increase information and awareness about general changes in eating and physical activity habits. - Provide general behavioural targets. - Allow children to express their readiness to change habits. - Assessing children’s stage of change (baseline stage).	No commitment to any change process.
Stage 2 Contemplation	Intention to change within the next six months. Aware of their problems. Interest in change behaviours but not ready to make changes yet. Plans to change without commitment.	Learning module 2 - Increase information and awareness about a specific behaviour related to the learning module topic. - Encourage children to discuss and solve barriers to healthy changes and express their own problems. - Help children to make clear the benefits and costs of changing. - Motivate children to set a specific behaviour goal.	Consciousness-raising: increasing information about the topic that supports a healthy change. Dramatic relief: expressing feelings about unhealthy behaviours related to problems and solutions. Decisional balance: pros and cons of changing.
Stage 3 Preparation	Intention to change soon (next month). Makes small changes. Has a plan to change.	Learning module 3 - Increase information and awareness about a specific behaviour related to the learning module’s topic. - Encourage children to discuss and solve barriers to healthy changes and express their own problems. - Help children to make clear the benefits and costs of changing. - Reinforce small changes if already achieved by children. - Encourage children to set a specific behaviour related to the topic’s module.	Consciousness-raising: increasing information about the topic that supports a healthy change. Dramatic relief: expressing feelings about unhealthy behaviours related to problems and solutions. Decisional balance: pros and cons of changing. Environmental reevaluation: the combination of cognitive and affective assessment to realise the impact of unhealthy behaviour on one’s family environment.
Stage 4 Action	Makes specific behaviour changes for less than six months. Visible behaviour changes.	Learning modules 4, 5, 6, 7 - Increase information and awareness about a specific behaviour related to the learning module’s topic. - Encourage children to discuss and solve barriers to healthy changes and express their own problems. - Help children to make clear the benefits and costs of changing. - Develop competencies for decision-making on healthy food and active life habits. - Reinforce small changes reported by children. - Support confidence. - Encourage children to set a specific behaviour related to the topic’s module.	Consciousness-raising: increasing information about the topic that supports a healthy change. Dramatic relief: expressing feelings about unhealthy behaviours related to problems and solutions. Decisional balance: pros and cons of changing. Self-liberation: making a solid commitment to change and decision-making abilities. Counterconditioning: learning healthier behaviours to substitute unhealthy ones. Self-efficacy: confidence that people follow healthy behaviours in challenging situations without relapsing to old habits.
Stage 5 Maintenance	Makes specific behaviour changes for six months or longer. Intention of preventing unhealthy behaviours. Increasingly confident of continuing changes.	Learning module 8 - Provide a recall of the overall modules. - Reinforce small changes reported by children. - Encourage children to continue to follow the healthy behaviours achieved. - Support confidence.	Self-liberation: making a solid commitment to change and decision-making abilities. Self-efficacy: confidence that people follow healthy behaviours in challenging situations without relapsing to old habits.

**Table 2 nutrients-15-04543-t002:** Process evaluation and data collection.

Process Evaluation/Method	Participants	Timeline	Measurements
Schools setting scan	School principals and teacher coordinators of the four schools	Phase 1	School conditions (semi-structured interview): - School curriculum; - Textbooks; - Physical education classes; - School food services.
Anthropometric assessment	Intervention group and control group	Phase 1 Baseline and Follow-up	Anthropometric measures: height, weight, waist circumference (WC). Calculation of: BMI, waist-to-height ratio (WHtR) and height, weight and BMI for age and sex-specific z-scores.
Self-reporting questionnaire	Intervention group and control group	Phase 1 Baseline and Follow-up	Socio-demographic and lifestyle behaviours. Age. Education level and occupation of parents. Food knowledge score. Physical activity, including sport participation. Sedentary behaviours: watching TV, playing computer and videogames. Food frequency questionnaire (FFQ).
List of topics for the learning modules	Intervention group	Phase 1 Baseline	List of 16 topics; 8 of them to be selected by children.
Food Records	Intervention group	Phase 2 Over the PHS-pro (between baseline and follow-up)	Measuring dietary change according to the multiple target behaviours of the PHS-pro: - Consumption of F&V; - Beverages contributing mostly to children’s daily diet (water, milk, yoghurt, soft drinks); - High-fat, high-sugar food and energy-dense food consumption.
Feedback of participants Children’s questionnaire	Intervention group	Phase 2 After PHS-pro	Children’s perceptions about: - PHS-pro components; - Eating behaviour changes.
Feedback of participants Teachers’ questionnaire and SWOT analysis (Strengths, Weaknesses, Opportunities and Threats analysis)	Teacher staff of intervention school	Stage 2 After PHS-pro	Teachers’ perceptions about: - PHS-pro components; - Teachers’ proposals for a better programme (SWOT analysis).
Field notes	All participants	Phase 1 Phase 2	Documenting and registration of activities, meetings, interviews, information, documents and time spent to conduct the research.

**Table 3 nutrients-15-04543-t003:** Children participants’ inclusion or exclusion.

Participant inclusion criteria: Sixth grade classes of elementary school; Written consent of parents to participate in the study.
Participant exclusion criteria: Not attending the two assessments, baseline and follow-up; Receiving monitoring or any nutritional intervention; Having pathological conditions, which require taking medication that might interfere with the body’s metabolism or be associated with weight changes; Being identified with special educational needs as declared by the teacher; Not wanting to participate in PHS-pro (for the intervention school).

**Table 4 nutrients-15-04543-t004:** Children’s stage of readiness to change their behaviours based on the “10 steps to be healthier”.

Stages of Change	Class A (*n* = 19) Step n°	Class B (*n* = 27) Step n°	Class C (*n* = 27) Step n°	Class D (*n* = 24) Step n°	Class E (*n* = 24) Step n°	Class F (*n* = 26) Step n°	Class G (*n* = 28) Step n°	Class H (*n* = 26) Step n°	Class I (*n* = 26) Step n°
Pre Contemplation	-	-	-	-	-	-	-	-	-
Contemplation	3 4 7 8	4 5 7 8	4 5 7 8	3 5 7 8	5 7 8	2 5 7 8	5 7 8	7 8	4 7 8
Preparation	2 5 10	2 3 10	2 3 10	2 4 10	2 3 4 10	3 4 10	2 3 4 10	2 3 4 5 10	2 3 5 10
Action Maintenance	1 6 9	1 6 9	1 6 9	1 6 9	1 6 9	1 6 9	1 6 9	1 6 9	1 6 9
“10 steps to be healthier” list (see Section 2.3.2)									
1. Daily breakfast intake					6. Two snacks midmorning and afternoon				
2. Eating vegetables soup to start lunch and dinner					7. Avoiding high rich fat food				
3. Eating a colourful diet					8. Reducing salt				
4. More fruit: three different fruits each day					9. Drink water				
5. Switches meals: balance between meat and fish					10. Regular physical activity and watching the weight				

**Table 5 nutrients-15-04543-t005:** Frequencies of the returned food records.

N° of Food RecordsCollected	Boys(*n* = 119)	Girls(*n* = 110)	Total(*n* = 229)	Total%	
0	11	1	12	5.2	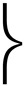 28%
1	8	7	15	6.6
2	15	7	22	9.6
3	10	5	15	6.6
4	15	7	22	9.6	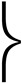 72%
5	15	16	31	13.5
6	17	25	42	18.3
7	28	42	70	30.6

**Table 6 nutrients-15-04543-t006:** Children’s views of their participation in the PHS-pro and their perceptions of their changing eating habits (*n* = 227).

Opinions	(*n* = 227) *n* (%)
**Q1.** *What were the two components you liked less* *during your participation in the PHS-pro? You can pick two boxes.*	
a. 58-item Food Frequency Questionnaire	13 (5.7%)
b. Anthropometric assessment	15 (6.6%)
c. Short story reading session	7 (3.1%)
d. Literary contest	11 (4.8%)
e. Learning modules	3 (1.3%)
f. Learning module topics	6 (2.6%)
**g. Food record**	**124 (54.6%)**
h. Anthropometric assessment information card	2 (0.9%)
**Perceptions**	
**Q2.** *Do you think your eating habits changed during the programme?*	
**Yes**	**192 (84.6%)**
No	35 (15.4%)
**Q3.** *If your answer is affirmative, would you like to explain how?*	
**Eating more fruit**	**78 (40.6%)**
Eating more vegetables or salads	41 (21.4%)
**Eating more vegetable soup**	**53 (27.6%)**
Eating more fish	13 (6.8%)
Drinking more milk	11 (5.7%)
Drinking more water	26 (13.5%)
Drinking less soft drinks	12 (6.3%)
Eating less sweets	38 (19.8%)
Eating less fried foods	33 (17.2%)
Eating breakfast	3 (1.6%)
**Q4.** *Do you think this kind of programme can help young people be more careful with food?*	
**Yes**	**223 (98.2%)**
No	4 (1.8%)
**Q5.** *If your answer is affirmative, would you like to explain why?*	
To know more about nutrition	8 (3.6%)
For preventing obesity	36 (16.1%)
**To help eat right**	**100 (45.0%)**
Eating for better health	16 (7.1%)
Greater awareness of food and nutrition	24 (11.0%)
It helps for better lifestyle habits	12 (5.4%)
For weight control	20 (9.0%)

**Table 7 nutrients-15-04543-t007:** Teachers’ evaluation questionnaire regarding the implementation (*n* = 16).

Components	Teachers (*n* = 12)
1. Organisation	6 inputs
	Extra-class organisation successful.
	Extra-class organisation went very well. No negative aspects.
	Extra-class organisation worked and was very helpful for general results.
	Teachers as key stakeholders need more coordination with more meetings to make a progress report. A wider programme dissemination would be more effective for the school community.
	Creation of a support group for coordinating participation and work of other stakeholders. More meetings for evaluation and for having a progress report.
2. Literary contest	9 inputs
	Contest criteria should be with more anticipation and clarification. Portuguese language teachers should have greater involvement as well as the library, which should have more participation.
	Good adhesion of students to the competition, from which emerged interesting works and could serve as suggestions for future initiatives.
	An enriching experience. It crossed the creative writing with the acquisition of new expressions and perceptions about eating habits. I was also able to observe that there was a commitment of the Portuguese language teachers, a determining factor for the quality of the tales. Having children as members of the jury, seemed to me, was an interesting idea, giving greater visibility to the issues. The prize-giving ceremony involving parents was also a very positive factor.
	Children showed enthusiasm in the elaboration of the stories.
	I could see that most of the children were enthusiastic and cooperated immediately.
	I think that the prize-giving ceremony should not happen on the last day of school but in the penultimate week. The jury and contest criteria were well chosen.
	The literary competition did not arise in the best way due to the unwillingness of some of the Portuguese teachers.
	The literary competition went well; children were very motivated.
	It was an important phase of the project, in which children were involved enthusiastically.
3. Learning modules	7 inputs
	Interesting topics, linked together and providing continuity. A reasonable period of time. Most of the children liked it.
	Relevant topics, clear and adequate. The possibility of choosing the topics by the children was a very positive approach.
	Interesting topics, good running time and good acceptance.
	Topics are quite important for food education and were well accepted by the children. They were always asking when the nutritionist returned to the class. Good learning of the 10 steps with a positive perception of behaviours. Modules had enough time.
	Modules were appropriate to the age group (language and graphics).
	Modules were appropriate to the age group.
	Modules have gone well.
	PowerPoint presentations can be improved.
4. Seven 3-day food records	6 inputs
	Children complained of being repetitive. A large number had failed to do so.
	Interesting dynamics, putting in evidence the reality of practices and behaviours, often “disguised”. Some resistance on the deliveries, perhaps because they were many.
	Interest and commitment in carrying out the registration daily; however some children gave up the task.
	In the beginning, the food record had good participation; however, by the end, some children got tired of the task, were reluctant and failed to deliver. The food record was well accepted by children and their parents.
	The food record was well accepted by the children but then there were withdrawals.

**Table 8 nutrients-15-04543-t008:** Results of teachers’ SWOT analyses about the components involved in the PHS-pro.

Strengths, Weaknesses, Opportunities and Threats Analysis (SWOT)	
Programme’s Overall Organisation	
Strengthens	Weaknesses
Possibility of children choosing the topics. Participatory methodology approach in different components (learning modules, literary contest jury) stimulating responsibility and meeting deadlines. Triggers creativity writing and teamwork linking with health and food topics. Organisation on time gave the possibility for the programme to be included in the school curricular project.	The lack of universal communication to all teachers to involve the school community. Difficulty in reaching some teachers and motivating them to participate. Limited communication with parents. Lack of providing training to the school community (teachers, kitchen staff, parents and those who want to improve their lifestyle habits).
School Environment	
Opportunities	Threats
Providing knowledge of the children’s nutritional status creates the possibility of changing the situation: preventing obesity from rising. Motivates children to adopt healthy habits. Supports healthy choices and a healthy environment in school. Possibility of creating extra-class dynamics with physical activity classes. Chance for reforming the school canteen and buffet and motivating healthy food choices. Possibility to bring healthy messages to parents. It may stimulate collaboration between stakeholders and the school community. May spread the programme format to other grades.	Resistance of children to delivering the food record form for behaviour change monitoring. The food record registration was seen as one more type of homework. Lack of a continuum of communication with parents over the intervention.

## Data Availability

The data presented in this study are available on request from the corresponding author. The data are not publicly available due to data privacy of the participants.

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
