# Peer review of "The “Planning Health in School” Programme (PHS-pro) to Improve Healthy Eating and Physical Activity: Design, Methodology, and Process Evaluation"

_nutrients, 2023, doi:10.3390/nu15214543_

Round 1
Reviewer 1 Report
Overall, it is a well-written document. Lots of detailed information is provided to the readers which enables reproducibility.
1. small typo mistakes need to be careful, some abbreviations have no full name, for example, SWOT.
2. If it is possible please add the cost-effective analysis in brief to indicate whether is it sustainable.
It is easy to read and follow.
Author Response
We sincerely thank the reviewers for careful reading of our manuscript and the constructive comments and suggestions, which allowed to help revising it. Our response follows the reviewer comments.
Review 1
Overall, it is a well-written document. Lots of detailed information is provided to the readers which enables reproducibility.
- small typo mistakes need to be careful, some abbreviations have no full name, for example, SWOT.
- If it is possible please add the cost-effective analysis in brief to indicate whether is it sustainable.
Response: Thank you for your comments that were extremely helpful and we have revised accordingly.
Following the suggestions, SWOT analysis has now full name for a clear reading. We also have revised other abbreviations.
Regarding the cost and benefits analysis (one of the four research objectives), they were already described in a published study (reference 41), however we added the most important finding in the final of section 1.2 (marked in underline-blue).

Reviewer 2 Report
I recommend using the CONSORT checklist to support the reporting of the protocol (https://eme.cochrane.org/consort-checklist-para-relatar-um-ensaio-clinico/). Obviously, not all the items can be answered, but in any case, it is an internationally recommended tool for qualifying the report.
I recommend improving the quality of Figure 1, it is difficult to read.
There is one point that I find interesting: is PHS-Pro included in the school curriculum so that, after study and possible reorganization, it can become a permanent action / public policy in schools?
Another point I recommend is the use of analyses with data imputation (e.g. intent-to-treat method) for children who eventually dropout / withdrawal of the study. These methods produce results with less risk of bias when compared to results based only on subjects who completed the study.
Author Response
We would like to thank you for your valuable comments concerning our manuscript. We have followed your recommendations, which we hope meet with your approval. We answer your comments in details in the following texts.
Review 2
I recommend using the CONSORT checklist to support the reporting of the protocol (https://eme.cochrane.org/consort-checklist-para-relatar-um-ensaio-clinico/). Obviously, not all the items can be answered, but in any case, it is an internationally recommended tool for qualifying the report.
Response: In fact, we followed the Consort checklist of the 22 items for the behavioural interventions, but since we could not answered to all the items, and our design followed a non-randomised trial, we considered that it might not be appropriate. As suggested by the reviewer, we added a sentence in the data collection section.
I recommend improving the quality of Figure 1, it is difficult to read.
Response: we have improved the quality of Fig 1.
There is one point that I find interesting: is PHS-Pro included in the school curriculum so that, after study and possible reorganization, it can become a permanent action / public policy in schools?
Response: yes, it could. Also, it can be adapted to the characteristics of each school environment. However, in Portugal, this area of prevention programmes to be activated, needs to be developed from the political point of view first. And, as far as we know, there is no financial budget to apply in schools.
Another point I recommend is the use of analyses with data imputation (e.g. intent-to-treat method) for children who eventually dropout / withdrawal of the study. These methods produce results with less risk of bias when compared to results based only on subjects who completed the study.
Response: As we referred in subsection 3.2.1, the participation rate of children in the PHS-pro school intervention was 99.1. Regarding the overall research and its 4 objectives, for each objective and respective study, we applied different strategies for the collected data. In this study, we only used the content analysis, for which we not considered the strategies of the intent-to-treat method.

Reviewer 3 Report
First of all, I would like to congratulate the authors for their research. As points of improvement:
1) Develop in a complete way the theoretical framework of the study. It is very scarce.
2) Add the research questions
3) Revise the English wording.
The article is very well developed. Statistically presented it is very well justified. I suggest that the above changes be implemented.
English should be revised
Author Response
We sincerely thank the reviewer for careful reading of our manuscript and the constructive comments and suggestions. Our response follows the reviewer comments.
Review 3
First of all, I would like to congratulate the authors for their research. As points of improvement:
- Develop in a complete way the theoretical framework of the study. It is very scarce.
2) Add the research questions
3) Revise the English wording.
The article is very well developed. Statistically presented it is very well justified. I suggest that the above changes be implemented.
Response: Thank you for your excellent observations. We strongly agree with the comments.
As suggested by the reviewer, we have changed the manuscript accordingly: the background was improved and it was included more references, as well as the theoretical framework section (marked in underline-blue). In subsection 2.2.2 and Table 1, it is already developed the approach of TTM for the programme. Research objectives were added in section 1.2 and we also rephrased the text. And we have revised the text for improving the English wording.
